# Active and Passive Carbon Fractions in Contrasting Cropping Systems, Tillage Practices, and Soil Types

S. Rakesh [1,*], Abhas Kumar Sinha [1], Deepranjan Sarkar [2], Dewali Roy [3], Divya Bodiga [4], Samaresh Sahoo [1], Prakash Kumar Jha [5], Pradeep Kumar Dubey [6] and Amitava Rakshit [7]

1    Department of Soil Science and Agricultural Chemistry, Uttar Banga Krishi Viswavidyalaya, Coochbehar 736165, India
2    Integral Institute of Agricultural Science and Technology, Integral University, Lucknow 226026, India
3    Division of Soil Science and Agricultural Chemistry, Indian Agricultural Research Institute, Pusa, New Delhi 110012, India
4    Department of Silviculture & Agroforestry, Sam Higginbottom University of Agricultural, Technology and Sciences, Prayagraj 211007, India
5    Feed the Future Sustainable Intensification Innovation Lab, Department of Agronomy, Kansas State University, Manhattan, KS 66506, USA
6    Institute of Environment and Sustainable Development, Banaras Hindu University, Varanasi 221005, India
7    Department of Soil Science and Agricultural Chemistry, Institute of Agricultural Sciences, Banaras Hindu University, Varanasi 221005, India
*    Correspondence: rakisavan.940@gmail.com

**Abstract:** The rate of change in the relative amount of active and passive carbon (AC and PC) due to the land management practices (cropping systems combined with tillage) may vary with soil types depending on their level of chemical and/or physical protection from the decomposition but has rarely been directly measured. We have quantified the C storage potentiality of different soil types, namely old alluvial Inceptisol of Malda and recent alluvial Entisol of Coochbehar in West Bengal (subtropical eastern India) under the influence of different cropping systems (rice-maize: RM and rice-wheat: RW) and tillage practices (zero-tillage: ZT and conventional tillage: CT). The key objective was to demonstrate the short-term impact of conservation agriculture (CA) on soil C dynamics over the conventional practice. Research revealed that after short-term CA, total organic carbon (TOC), AC, PC, and total nitrogen (TN) showed significant ($p < 0.05$) improvement under the RM cropping system over the RW. The highest TOC content under the RM cropping system was recorded in the sites of Malda over the Coochbehar sites. The ZT significantly ($p < 0.05$) enhanced the TOC in the upper layers (0–5 and 5–10 cm) and the CT showed improvements in the lower depths (10–20 cm). We observed some irregular variations in the interactions of the cropping system and tillage with respect to different sites. However, the ZT performed better in improving C fractions under RM and RW as compared to CT. The TOC and TN stocks were maximum in the lower depth which was evident in both soil types. The TOC linearly regressed on TN accounted for 94.2% variability ($R^2 = 0.942$) of the C accumulation in soil and vice-versa. The PC was in a significant relationship with TN ($R^2 = 0.943$), but AC was moderately regressed ($R^2 = 0.851$). Lower stratification ratio values in Coochbehar soils (sandy loam in texture) indicated higher profile distribution of AC and PC in the soil profile; while in the Inceptisol, accumulation of the C fractions on the soil surface due to heavy texture resulted in the higher stratification values. The novelty of this study is that old alluvial Inceptisol showed a comparatively greater amount of AC and PC storage capability in comparison with the new alluvial Entisol. Conclusively, our study demonstrated that the adoption of conservation agriculture (CA practice/ZT) in cropping systems with higher C biomass input would significantly enhance the AC and PC fractions; however, the amount of storage is highly governed by the soil type and climatic factors.

**Keywords:** C fractions; zero-tillage; rice-wheat system; rice-maize system; Entisol; Inceptisol

## 1. Introduction

Conservation agriculture (CA)/no-till farming-based management practices involving crop residue retention and minimum soil disturbance have been widely practiced and have ensured food security in South Asia [1,2]. This farming technique was recommended by the Voluntary Guidelines for Sustainable Soil Management, published by the United Nations Food and Agriculture Organization (UNFAO) in 2017 [3]. Because of its economic and environmental benefits, the no-till system has been a subject of extensive research and increasing popularity. Escalating carbon dioxide ($CO_2$) concentrations in the earth's atmosphere are being a serious concern in the present environmental scenario [4]. Thus, sequestrating $CO_2$ in terrestrial ecosystems has spurred interest among researchers [5,6]. As per the estimations made by the Intergovernmental Panel on Climate Change (IPCC), sectors such as agriculture, forestry, and/or other land uses together contribute about 22% of global anthropogenic greenhouse gas (GHG) emissions [7]. A significant amount of loss of soil organic matter (SOM) has occurred by the intensive tillage (or practices) that emits $CO_2$ into the atmosphere [8,9]. Capturing this $CO_2$ and storing it in the soil profile as soil organic carbon (SOC) would benefit both the soil and the environment. Approximately, about 1200 to 1600 Pg (1 Pg = 1015 g) of SOC pool and 695 to 930 Pg of inorganic carbon are present in the top 1 m soil [10]. The concentration of atmospheric $CO_2$ may be greatly impacted by the changes in the SOC pool, thus affecting the global carbon cycle [11]. Therefore, it is critical to maintain and restore SOC to address issues such as food security and climate change. Several studies demonstrated that the adoption of no-till farming provides adequate crop yields [12–15] through improved SOC content [16], biological activity [17] and soil aggregate stability [18]. However, there is meagre information available on how active and passive carbon fractions behave under varying cropping systems and tillage practices in different soil types.

Profile C sequestration and the stratification ratio are the key indicators of soil quality [19]. Studying these helps in understanding the layer-wise C accumulation behavior of soil and also in identifying the variations in the quality of topsoil SOM [20]. The SOM is the end product of dead plant residues, humus carbon, particulate organic carbon, and recalcitrant carbon, playing its pivotal role in soil quality, crop production, and environmental functions [21]. The SOM is also associated with soil aggregation and nutrient cycling, and also acts as an energy and physical habitat for microbial activity [22]. The status of SOM is often projected by determining the SOC content of the soil. Changes in SOC are influenced by management practices, fertilization, and the carbon (C) biomass addition to the field [23].

To understand if SOC is stored and/or lost through the soil, studying the SOC fractions that have varying residence time, viz., the labile and non-labile pool, is important. Usually, the labile fractions are highly active, rapidly decomposed, and most sensitive to fluctuations driven by environmental factors [24]. While, the non-labile pool (passive pool) is more stable, resistant to decomposition, and generally occurs as organic-mineral complexes. However, it decomposes slowly to microbial activities [25]. There has been a growing interest to study the SOC fractions as it is useful in classifying several types or fractions of SOC such as active C (AC) or potassium permanganate ($KMnO_4$) extractable C and recalcitrant or passive C (PC) with various residence or turnover times. Labile C fractions can have a direct impact on the microbial activities and C dynamics in the soil and these are the reflection of the soil ecosystem. These are a relatively smaller fraction of total organic carbon (TOC) with a very short half-life in soils [26]. These C pools provide energy for soil microbes, which determine how biologically fertile the soil is. Additionally, changes in rainfall patterns may influence soil microbial activities positively or negatively [27], thus altering litter decomposition, SOC mineralization, and SOC pools [28]. Assessment of labile C fractions aids in the provision of determining the short-term changes in the quality of agricultural land [29–31].

Tillage and crop residue retention affect the SOM through the manipulation of organic matter input resulting in higher addition of OM into the soil under zero-tillage (ZT) when

compared to the conventional tillage (CT) practice [32]. Frequent cultivation of arable lands leads to a large loss of $CO_2$ from soil to the atmosphere which results in low SOM in the soil profile. Changes in soil carbon storage/carbon stock need a longer period to occur which is also very crucial to distinguish the active and passive SOC fractions from the SOC pool when determining the influence of varying agricultural management practices on soil carbon dynamics [33,34].

Very limited research suggests that the farming system is the main cause of any change in carbon storage. For example, the rice–wheat cropping system had much less effect on SOC as compared to the wheat–maize system in slightly alkaline soils [35]; although less is known about such impacts in the Eastern Gangetic Alluvial Plains. In the Indo-Gangetic Plains (IGPs) of South Asia, the rice-wheat cropping system is dominated by conventional management practices that include intensive tillage, non-judicious fertilizer management, less application of organic manures, etc. [36,37]. As a consequence, there is soil degradation, declined soil productivity, low crop yields, and degradation of the natural resource base [38,39]. Carbon storage in the soil profile helps in enhancing soil aggregation, restoring degraded lands, increasing soil fertility, and overall crop productivity [40–42]. Therefore, prevailing agricultural production in these regions with cost-effective technologies such as conservation agriculture involved with ZT, crop residue mulching, and crop diversification helps in achieving agricultural and environmental sustainability [43,44].

To date, there is plentiful research that investigated the effect of various crop management practices on soil properties and crop productivity [45–47]. However, literature on the interaction effect of cropping systems and tillage on active and passive C fractions at different depths, specifically in the surface soils of Indo-Gangetic Plains (IGP), is limited. The novelty of the present research is the comparison of the C fractions storage capability of different soil types under varying environments as influenced by the combined effect of cropping systems and tillage practices.

Our study hypothesized that an alteration in the cropping systems and tillage techniques has a differential impact on the status and storage of the active and passive C pools at different soil depths. Thus, it is pivotal to evaluate the current practices on the SOC fractions so that multiple goals of sustaining high crop yields and efficient resource utilization can be matched. In this background, the objectives of the present investigation included (i) assessing the response of active and passive C fractions to varying cropping systems and tillage practices at different soil depths over the period of 4 years; (ii) exploring the stratification of active and passive C fractions in the soil profiles of different agroecosystems; and (iii) determining the relationships between active and passive C fractions and soil properties under different agroclimatic conditions.

## 2. Materials and Methods

### 2.1. Site Description

The present study was undertaken in the experimental fields of Coochbehar (26.3452° N, 89.4482° E) and Malda (25.0108° N, 88.1411° E) districts of West Bengal state, India, of the Australian-funded (Australian Centre for International Agriculture Research, ACIAR) project Sustainable and Resilient Farming System Intensification (SRFSI) initiated in 2013 to demonstrate the benefits of CA practice over the conventional practice.

In the northern parts of West Bengal, especially in Malda and Coochbehar districts, there are different types of soils and climates. Malda soils belong to the soil order "*old alluvial Inceptisol*" and Coochbehar soils belong to "*new alluvial Entisol*". Farming in these parts is facing the same challenges (depletion of SOM and groundwater, multi-nutrient deficiencies, the rising cost of inputs, etc.) as other parts of the world and has adapted the improved and modern resource conservation techniques for better production and productivity of the soil. In total, there were seven field experimental sites ((three in Coochbehar (excluded 1 site due to fundamental errors) and four in Malda) involving two cropping systems (RW and RM) and two tillage practices (ZT and CT) with three replications in a factorial design (2 × 2).

Coochbehar receives an average annual rainfall of 2357 mm and Malda receives 1358 mm. The maximum and minimum temperatures in Coochbehar are 28.2 °C and 20.0 °C; in Malda, 30.6 °C and 20.2 °C, respectively. The daily weather parameters (daily temperatures and rainfall) of both districts during the experimental years (June 2014 to May 2018) have been illustrated in Figure 1. The total rainy days were much higher in Coochbehar with some extremely high rainfall events (up to 250 mm in a single day) than in Malda (Figure 1). Coochbehar soils are slightly sandy textured, acidic in nature, and are recently deposited alluvium soils [15,48]. Malda soils are entirely different from Coochbehar. Soils are silty to clayey in texture with neutral to alkaline in nature and are an old alluvial material. The salient features of these two types of soils are presented in Supplementary Table S1.

### 2.2. Treatment and Sampling Details

The present experiment and the fields were selected under the aegis of the SRFSI project, which was maintained by the Uttar Banga Krishi Vishwavidyalaya (UBKV), West Bengal. The field sites (seven sites of two different districts) selected for this study were historically used for growing rice in rotation with other dry-season crops using intensive tillage practices. Such fields were selected by the SRFSI to improve farming livelihoods.

Each cropping system, i.e., RW and RM, comprised of two tillage treatments, i.e., ZT: unpuddled transplanted rice (UPTR)—ZT maize or wheat and CT: Puddled transplanted rice (PTR)—CT maize or wheat. There were 3 farmers from each of 2 cropping systems, i.e., RM and RW which consisted of 2 tillage systems, sampled at 3 soil depths in the 7 sites altogether making 252 samples.

Initial (in 2013) soil samples were collected at 0–20 cm depth to estimate the initial physicochemical properties of the soils. After harvesting the wheat crop in April 2017–18 and maize in May 2017–18, soil samples were collected at three different depths, i.e., 0–5, 5–10, and 10–20 cm from each plot. Samples were then air-dried and ground to pass through a 2 mm sieve for chemical analysis and a 0.5 mm sieve for estimating TOC, AC, and PC fractions.

### 2.3. Crop Management

Wheat and maize crops were sown immediately after the harvest of rice crops to capture the residual soil moisture. However, the date of sowing and harvesting varied across the field sites (FS) and the districts. The area of the experimental field was about 666 m$^2$ (0.07 ha). In the case of ZT, about 3.0 t ha$^{-1}$ rice residue was carried over in succeeding winter crops, and about 2.0 t ha$^{-1}$ and 5.0 t ha$^{-1}$ from wheat and maize, respectively, crop residue was retained while rice transplanting. In total, 5.0 t ha$^{-1}$ and 8.0 t ha$^{-1}$ of crop residue were recycled in the RW system and RM system, respectively, by keeping them as mulch. In the CT system, about 3.0 t ha$^{-1}$ residue in RW and 4.0 t ha$^{-1}$ residue in RM were incorporated annually during land preparation.

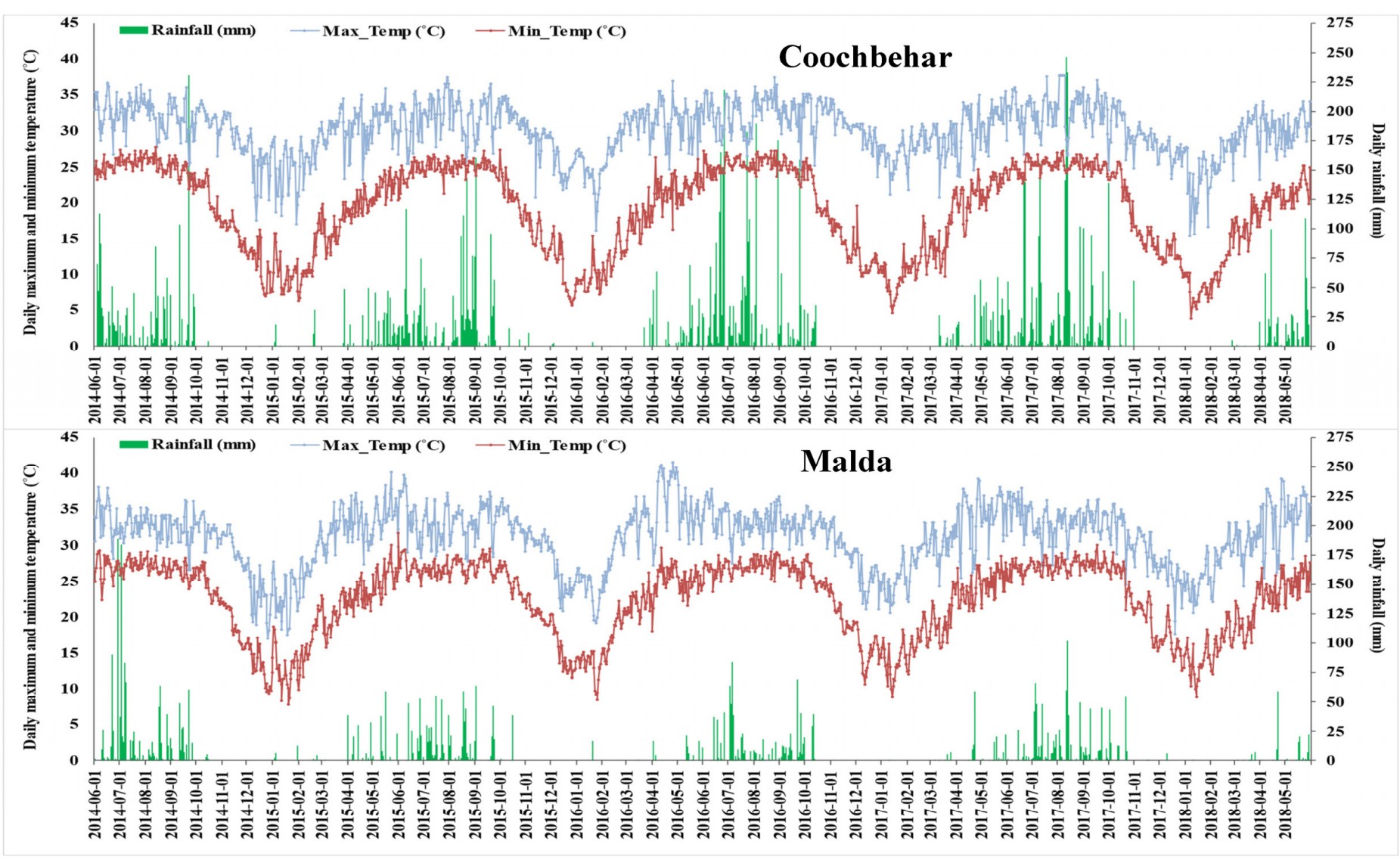

**Figure 1.** Weather parameters (daily temperatures and rainfall) of both districts during the experimental years (June 2014 to May 2018).

The tillage and cropping systems used for CT were: Puddled transplanted rice (PTR)–CT maize or wheat; and ZT: Unpuddled transplanted rice (UPTR)–ZT maize or wheat. Seedlings of rice were transplanted at 24 × 14 cm (row × plant) spacing in the ZT using a mechanical transplanter and planted randomly by hand in the CT resulting in 28–30 hills $m^{-2}$. Wheat was sown at 20 cm row spacing in the ZT with continuous seeding and broadcast in the CT with the seed rate of 120 kg $ha^{-1}$. Maize was planted at 60 × 20 cm (row × plant) in both the ZT and CT resulting in about 80000 plants $ha^{-1}$. Crops were fertilized at rates (kg $ha^{-1}$) recommended for the area: rice 80–90 N, 15–20 P, 40–70 K; wheat 125–145 N, 20–25 P, 40–60 K; and maize 155–180 N, 20–25 P, 60–75 K. Other agronomic practices are briefly discussed in the article published by Islam et al. [1].

*2.4. Soil Analysis*

2.4.1. Soil Properties

Soil pH was determined by a pH meter as described by Jackson [49]. Soil bulk density was estimated by following the method of Cresswell and Hamilton [50]. Soil texture was determined by the Bouyoucos hydrometer method [51]. The soil total nitrogen was analyzed by the Kjeldahl method [52].

2.4.2. TOC and its Fractions

TOC in soil was estimated by the colorimetric method following the modified Walkley and Black method [53] by taking exactly 1 g of soil and digesting it in 20 mL of 5% $K_2Cr_2O_7$ and 10 mL of concentrated $H_2SO_4$. Then, 50 mL of 0.4% $BaCl_2$ was added after cooling for 30 min and then allowed to stand overnight. The intensity of the yellow/orange color was read at 600 nm wavelength using a UV-visible spectrophotometer. Active carbon (AC), i.e., 0.2 M of potassium permanganate oxidizable carbon (POXC), was determined by the colorimetric analysis. About 2.5 g of soil was extracted with 2 mL of $KMnO_4$ solution and 18 mL of double distilled water (DDW) for 2 min. Then the samples were allowed to settle for 10 min in dark conditions. About 0.5 mL of aliquots were pipetted into 50 mL volumetric flasks and the rest of the volume was filled with DDW. Finally, the diluted samples (purple supernatant) were read at 550 nm wavelength [54]. Passive carbon (PC) concentration in the soil samples was computed by subtracting the values of active carbon from total organic carbon [55] as shown below:

Passive carbon (g $kg^{-1}$ soil) = Total organic C (g $kg^{-1}$) − Active carbon (g $kg^{-1}$).

*2.5. Computation of Stratification Ratio (SR)*

This ratio is the value at the surface layer to that at a lower depth [56]. For example, the value of 0–5 cm is divided by the value of 5–10 cm depth for a C fraction ratio at 0–10 cm depth. While for 0–20 cm, it is the value of 0–5 cm depth divided by the value of 10–20 cm soil depth.

*2.6. Computation of TOC/TN Stock*

Stocks were calculated by considering the concentration of the SOC or TN, bulk density, and soil depth using the formula shown below [57]:

$$\text{TOC/TN stock (Mg ha}^{-1}) = 10^4 \, \text{ha}^{-1} \times \text{BD (Mg m}^{-3}) \times \text{soil depth (m)} \times \text{TOC/TN (\%)} \times 10^{-2}.$$

*2.7. Data Analysis*

A factorial randomized completely blocked design was employed considering the two factors, i.e., cropping system and tillage in the fit-ANOVA model at $p < 0.05$ with separation of means by LSD using the SPSS 17.0 software package. A regression analysis of TN with AC and PC was performed using Microsoft excel 2016.

## 3. Results

### 3.1. Total Organic Carbon and Total Nitrogen

After 5 years of continuous (short-term) conservation agriculture, TOC and its fractions and TN showed some peculiar variations with reference to the management practices among the different sites. Almost all the sites of both districts revealed a significant ($p < 0.05$) improvement in the TOC concentration under the RM cropping system over RW (Table 1) except in site seven. The highest TOC content (17.13 g kg$^{-1}$) was observed in site four of Malda (*old alluvial Inceptisol*) and a value of 13.60 g kg$^{-1}$ was recorded in site five of Coochbehar (*new alluvial Entisol*) under the RM cropping system. With respect to the tillage system, ZT enhanced the TOC over CT in the four sites and a maximum value of 15.93 and 15.73 g kg$^{-1}$ was observed in site four under ZT and CT, respectively. In the rest of the sites, CT improved the TOC content in comparison to ZT, but in site three of Malda, we noticed a higher change in TOC (12.48 g kg$^{-1}$) under CT which was 10.54% higher than ZT (11.29 g kg$^{-1}$).

Depth-wise status of TOC resulted in a significant decrement in concentration with increasing depth (Table 1) among all the sites. However, greater amounts were recorded at 0–5 cm and relatively similar amounts of TOC were observed in 5–10 and 10–20 cm depths. We have statistically performed the significant interaction of CS × T, CS × D, T × D, and CS × T × D but in this paper, we focused and briefly discussed the interaction of CS × T. Our study of focus is on the main effects of CS and T and their interaction (CS × T). Other interaction tables (CS × D, T × D, and CS × T × D) are presented in the Supplementary Tables S2–S5.

The interplay between the cropping systems and tillage (CS × T) on TOC concentration showed a significant difference ($p < 0.05$) in almost all the sites except in site two of Malda and site seven of Coochbehar (Table 2). We observed some irregular outcomes from CS × T with respect to different sites. The ZT management enhanced the TOC in site one, site four, and site six under the RW system, and in site one, site five, and site six under the RM system as against CT (Figure 2a). Interestingly, in site four of Malda where TOC was higher compared to the rest of the sites, CT performed better under RM and ZT performed better under RW in improving the TOC. Overall, the performance of ZT under the RM system was better.

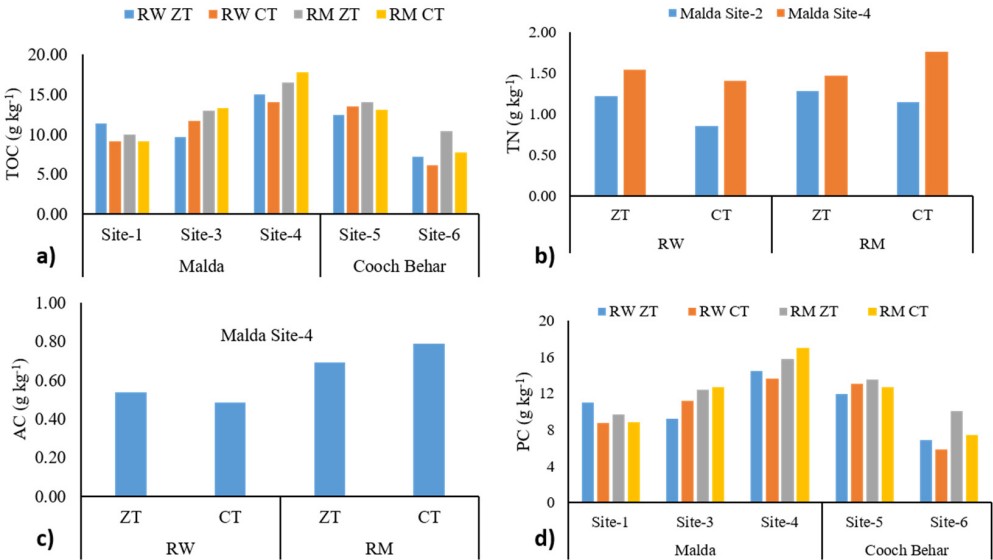

**Figure 2.** Significant ($p < 0.05$) interaction of cropping system and tillage (CS × T) on TOC (**a**), TN (**b**), AC (**c**), and PC (**d**).

**Table 1.** TOC, TN, AC, and PC as influenced by cropping system, tillage, and depth at different sites of Malda and Coochbehar.

| Parameter | Districts | | | Malda | | | | Coochbehar | | |
|---|---|---|---|---|---|---|---|---|---|---|
| | Location | | Site-1 | Site-2 | Site-3 | Site-4 | Site-5 | Site-6 | Site-7 |
| | CS | RW | 10.24 | 11.40 | 10.64 | 14.54 | 12.96 | 6.63 | 12.40 |
| | | RM | 9.57 | 13.13 | 13.12 | 17.13 | 13.60 | 9.05 | 12.51 |
| | | ($p < 0.05$) | 0.008 | 0.000 | 0.000 | 0.000 | 0.014 | 0.000 | NS |
| TOC | T | ZT | 10.67 | 12.76 | 11.29 | 15.73 | 13.24 | 8.74 | 13.49 |
| ($g\ kg^{-1}$) | | CT | 9.15 | 11.76 | 12.48 | 15.93 | 13.32 | 6.94 | 11.43 |
| | | ($p < 0.05$) | 0.000 | 0.009 | 0.001 | NS | NS | 0.000 | 0.000 |
| | D | 0 to 5 | 12.45 | 14.80 | 15.39 | 18.74 | 15.56 | 8.37 | 14.31 |
| | | 5 to 10 | 10.12 | 11.72 | 10.97 | 15.72 | 14.24 | 7.95 | 12.43 |
| | | 10 to 20 | 7.15 | 10.27 | 9.29 | 13.04 | 10.04 | 7.20 | 10.64 |
| | | ($p < 0.05$) | 0.000 | 0.000 | 0.000 | 0.000 | 0.000 | 0.000 | 0.000 |
| | CS | RW | 1.01 | 1.04 | 1.15 | 1.48 | 1.31 | 0.95 | 1.38 |
| | | RM | 1.20 | 1.21 | 1.36 | 1.62 | 1.45 | 1.03 | 1.42 |
| | | ($p < 0.05$) | 0.000 | 0.000 | 0.001 | 0.020 | 0.001 | NS | NS |
| TN | T | ZT | 1.08 | 1.25 | 1.29 | 1.51 | 1.35 | 1.05 | 1.42 |
| ($g\ kg^{-1}$) | | CT | 1.13 | 1.00 | 1.22 | 1.59 | 1.41 | 0.93 | 1.38 |
| | | ($p < 0.05$) | NS | 0.000 | NS | NS | NS | 0.028 | NS |
| | D | 0 to 5 | 1.26 | 1.32 | 1.47 | 1.89 | 1.63 | 1.16 | 1.53 |
| | | 5 to 10 | 1.19 | 1.15 | 1.32 | 1.44 | 1.37 | 1.04 | 1.45 |
| | | 10 to 20 | 0.86 | 0.90 | 0.98 | 1.31 | 1.14 | 0.78 | 1.21 |
| | | ($p < 0.05$) | 0.000 | 0.000 | 0.000 | 0.000 | 0.000 | 0.000 | 0.000 |
| | CS | RW | 0.37 | 0.41 | 0.43 | 0.51 | 0.46 | 0.30 | 0.32 |
| | | RM | 0.33 | 0.44 | 0.62 | 0.74 | 0.47 | 0.30 | 0.32 |
| | | ($p < 0.05$) | 0.000 | 0.005 | 0.000 | 0.000 | NS | NS | NS |
| AC | T | ZT | 0.37 | 0.47 | 0.49 | 0.61 | 0.50 | 0.30 | 0.37 |
| ($g\ kg^{-1}$) | | CT | 0.33 | 0.38 | 0.56 | 0.64 | 0.43 | 0.30 | 0.27 |
| | | ($p < 0.05$) | 0.001 | 0.000 | 0.000 | 0.015 | 0.000 | NS | 0.000 |
| | D | 0 to 5 | 0.53 | 0.54 | 0.72 | 0.78 | 0.56 | 0.33 | 0.40 |
| | | 5 to 10 | 0.30 | 0.38 | 0.44 | 0.60 | 0.49 | 0.28 | 0.28 |
| | | 10 to 20 | 0.22 | 0.34 | 0.41 | 0.50 | 0.34 | 0.30 | 0.28 |
| | | ($p < 0.05$) | 0.000 | 0.000 | 0.000 | 0.000 | 0.000 | 0.003 | 0.000 |
| | CS | RW | 9.87 | 10.99 | 10.21 | 14.03 | 12.51 | 6.33 | 12.09 |
| | | RM | 9.24 | 12.69 | 12.50 | 16.39 | 13.13 | 8.74 | 12.19 |
| | | ($p < 0.05$) | 0.014 | 0.000 | 0.000 | 0.000 | 0.014 | 0.000 | NS |
| PC | T | ZT | 10.30 | 12.30 | 10.80 | 15.11 | 12.75 | 8.44 | 13.12 |
| ($g\ kg^{-1}$) | | CT | 8.81 | 11.39 | 11.91 | 15.30 | 12.89 | 6.64 | 11.16 |
| | | ($p < 0.05$) | 0.000 | 0.014 | 0.001 | NS | NS | 0.000 | 0.000 |
| | D | 0 to 5 | 11.92 | 14.25 | 14.67 | 17.96 | 15.00 | 8.04 | 13.90 |
| | | 5 to 10 | 9.82 | 11.34 | 10.53 | 15.12 | 13.75 | 7.67 | 12.15 |
| | | 10 to 20 | 6.93 | 9.93 | 8.88 | 12.54 | 9.70 | 6.90 | 10.36 |
| | | ($p < 0.05$) | 0.000 | 0.000 | 0.000 | 0.000 | 0.000 | 0.000 | 0.000 |

Note: TOC—total organic carbon; TN—total nitrogen; AC—active carbon; PC—passive carbon; CS—cropping system; RW—rice-wheat; RM-rice-maize; T—tillage; ZT—zero-tillage; CT—conventional tillage; D—depth; NS—non-significant.

Similar to the TOC, the RM system enhanced the TN content significantly over RW (Table 1) in all the sites of Malda, but in Coochbehar it was evident only in site five. The maximum TN content was (1.62 g kg$^{-1}$) noticed in site four of Malda followed by the value 1.45 g kg$^{-1}$ recorded in site five of Coochbehar under the RM system and the least amount (0.95 g kg$^{-1}$) was recorded under RW in the site six. The tillage practices failed to show a significant effect on TN in the majority of the sites. Only site two and site six showed a 19.84% and 10.87% increase in TN, respectively, under ZT over CT (Table 1). The depth-wise concentration of TN followed a similar pattern as observed in TOC. The interplay between CS and T was not significant in any of the Coochbehar sites; however,

in Malda, site two and site four showed a significant improvement in TN under varied cropping systems (Figure 2b). In site two, ZT enhanced the TN under both RW and RM but in site four, ZT under RW and CT under RM improved the same.

**Table 2.** *p* values (<0.05) showing the significance of the interaction effect of cropping system and tillage (CS × T) on TOC, TN, AC, and PC at different sites of Malda and Coochbehar.

| Parameter | Malda | | | | Coochbehar | | |
|---|---|---|---|---|---|---|---|
| | Site-1 | Site-2 | Site-3 | Site-4 | Site-5 | Site-6 | Site-7 |
| TOC | 0.007 | NS | 0.011 | 0.011 | 0 | 0 | NS |
| TN | NS | 0 | NS | 0.001 | NS | NS | NS |
| AC | NS | NS | NS | 0 | NS | NS | NS |
| PC | 0.007 | NS | 0.009 | 0.015 | 0 | 0 | NS |

Note: TOC—total organic carbon; TN—total nitrogen; AC—active carbon; PC—passive carbon; NS—non-significant.

### 3.2. Active Carbon and Passive Carbon

The sites of Coochbehar did not show any significant change in AC under the cropping system effect, but in Malda, there was significant ($p < 0.05$) improvement among all the sites (Table 1). We have observed a 44% increment of AC in site three and site four under RM (0.62 and 0.74 g kg$^{-1}$, respectively) over RW (0.43 and 0.51 g kg$^{-1}$, respectively). ZT significantly improved the status of AC in all the sites except site three and site four of Malda. These two sites also resulted in higher values of AC under CT (0.56 and 0.64 g kg$^{-1}$ in site three and site four, respectively). The concentration of AC decreased with soil depth; the topmost layer (0–5 cm) recorded maximum AC content, and in the subsequent depths (5–10 and 10–20 cm), no variations were observed among all the sites. With respect to the interaction of CS × T, only site four showed a significant difference (Table 2) which revealed that ZT under RW and CT under RM enhanced the AC status (Figure 2c).

The PC was observed to be significantly ($p < 0.05$) improved under the RM system evident in all the sites of Coochbehar and Malda (Table 1). The highest value of PC (16.39 g kg$^{-1}$) was observed in site four of Malda followed by the value 13.13 g kg$^{-1}$ noted in site five of Coochbehar under RM. The least amount of PC (6.33 g kg$^{-1}$) was recorded in site six under the RW system. The ZT system significantly enhanced the status of PC in site one (14.3%) and site two (7.4%) of Malda; while in Coochbehar, it was in site six (21.3%) and site seven (14.9%) when compared with CT Management. The concentration and distribution of PC in the three soil depths (0–5, 5–10, and 10–20 cm) varied substantially in all the studied sites. It varied from 8.04 to 17.96 g kg$^{-1}$, 7.67 to 15.12 g kg$^{-1}$, and 6.90 to 12.54 g kg$^{-1}$ at 0–5, 5–10, and 10–20 cm, respectively. Unlike AC, the variation in PC was significant in the majority of the sites due to the interplay between the CS × T except in site two and site seven (Table 2). The variations observed for PC under the influence of CS × T followed a similar trend as that of TOC (Figure 2d).

### 3.3. Bulk Density, TOC Stock, and TN Stock

We noticed the highest bulk density in the soils of Malda (site one to site four), which varied from 1.30 to 1.47 g cc$^{-1}$, and in Coochbehar soil; it varied from 1.06 to 1.23 g cc$^{-1}$ (Figure 3). With the increase in soil depth, BD values increased, and it was evident in all the sites. However, higher BD was recorded in site four followed by site one of Malda, and the least value was recorded in site five of Coochbehar.

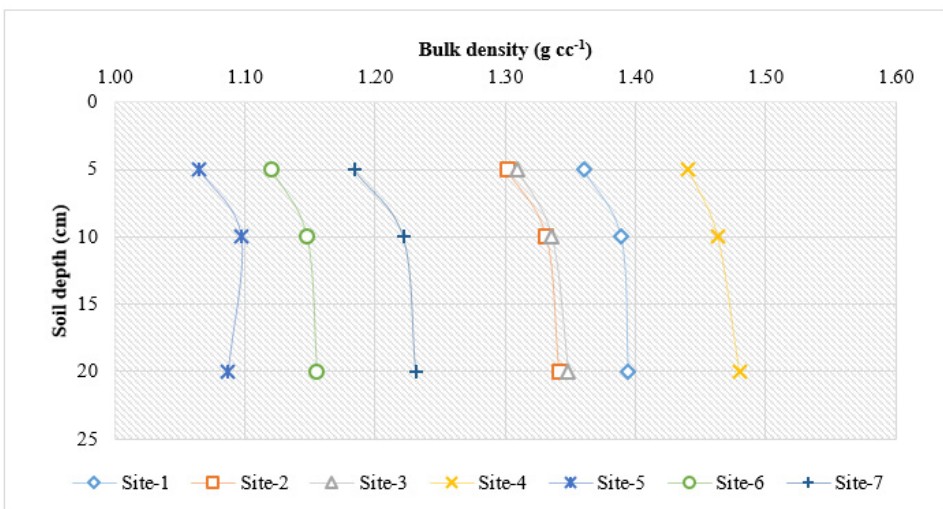

**Figure 3.** Depth-wise (0–20 cm) bulk density status of different sites (site-1 to 4: Malda; site-5 to 7: Coochbehar).

The TOC stock varied from 4 to 16 Mg ha$^{-1}$ and the TN stock varied from 0.6 to 1.4 Mg ha$^{-1}$ (Figure 4). The effect of the cropping systems (Figure 4a) significantly influenced the TOC and TN stocks in both districts. While the tillage practices (Figure 4b) showed significant effects on TOC stocks in both districts, but the effect of tillage on the TN stock was non-significant in all the sites of the Coochbehar (Figure 4). The RM system improved the TOC and TN stocks compared to RW. While, under two tillage practices, ZT enhanced both in sites one, two, six, and seven, and in the rest of the sites, CT dominated. The concentration of TOC and TN was also reflected in their stock values. Interestingly, under the CS effect, TN stock varied parallel with TOC stock, but under the tillage system, it varied unevenly in sites one, two, three, and seven where we observed higher TOC stock but lower TN stock and vice versa. Stock values of TOC and TN were observed to be maximum in the lower depth (10–20 cm) and lowest in the middle layer (5–10 cm) (Figure 5); however, the topmost depth (0–5 cm) was enriched with TOC and TN stock but lower than the lowermost evident in all the sites.

*3.4. Relationship of TOC, AC, and PC with TN*

When the TOC, AC, and PC contents over TN were plotted, we observed a strong positive correlation among them (Figure 6). TOC linearly and significantly regressed on TN, accounting for 94.2% of the variability ($R^2$ = 0.942) of the C accumulation in soil and vice versa. The PC also showed a similar significant relationship with 94.3% variability ($R^2$ = 0.943) but comparatively, AC showed slightly less variability, i.e., 85% ($R^2$ = 0.851) which moderately regressed on the TN content.

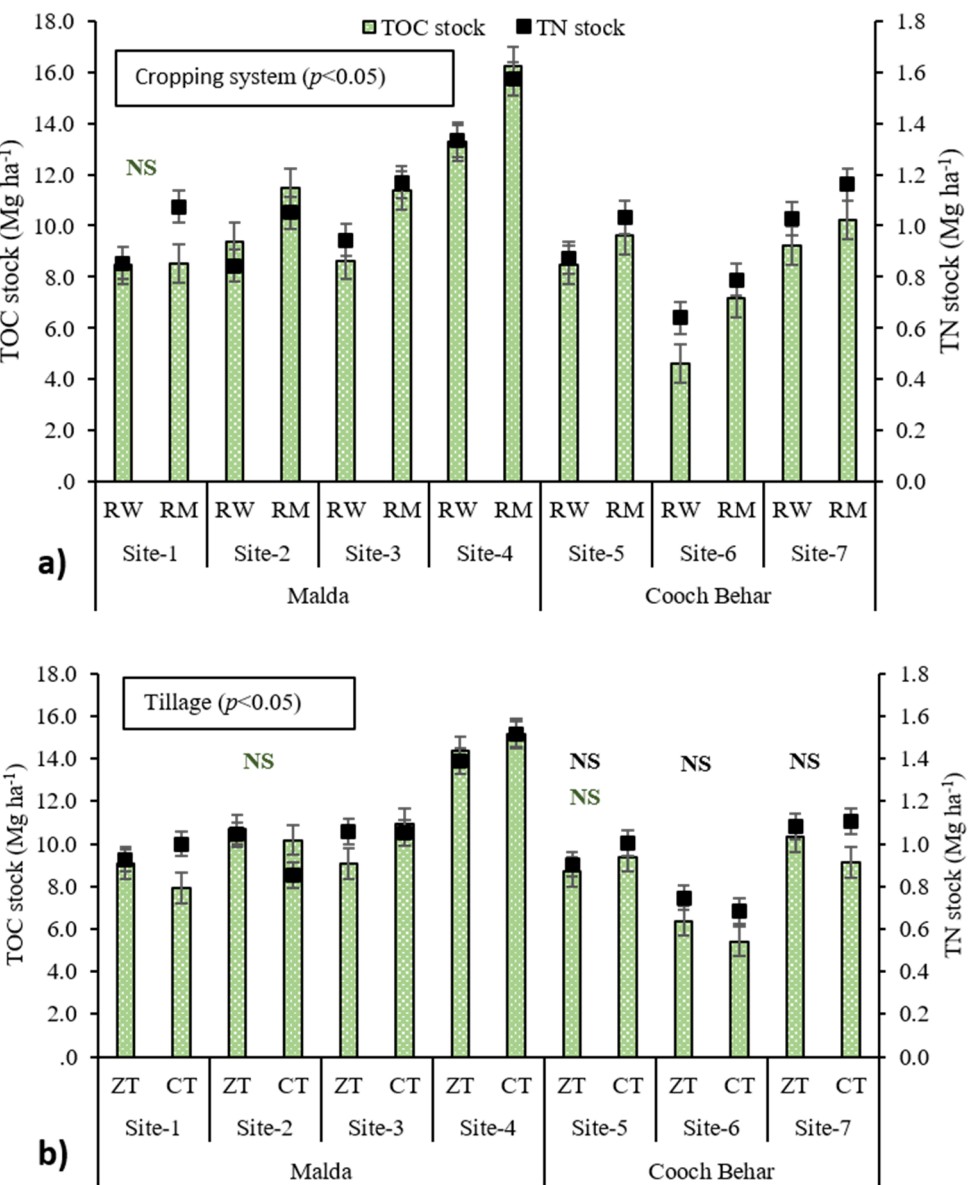

**Figure 4.** Effect of (**a**) cropping system (rice-wheat-RW; rice-maize-RM) and (**b**) tillage (zero-tillage-ZT; conventional tillage-CT) on TOC and TN stocks at different sites of Malda and Coochbehar. NS-non-significant; bars on the column/marker indicating standard error mean.

### 3.5. Stratification of AC and PC

The stratification ratio (Figure 7) of AC and PC revealed higher ratio values under the ZT system than the CT system, indicating a maximum accumulation in the upper layers among all the sites studied. The ratio values of 0–5/5–10 were observed to be less than 0–5/10–20, which means the concentration of accumulation between 0–10 cm was relatively higher than 0–20 cm. However, in Coochbehar soils (sites five, six, and seven), comparatively lower stratification ratio values were recorded which showed a higher distribution of AC and PC at 0–20 cm soil depth because the sandy texture of those soils enhanced the movement of C fractions into the lower depths appropriately. Lower ratio values under the CT system indicated a thorough distribution of C fractions due to the incorporation effect during tillage.

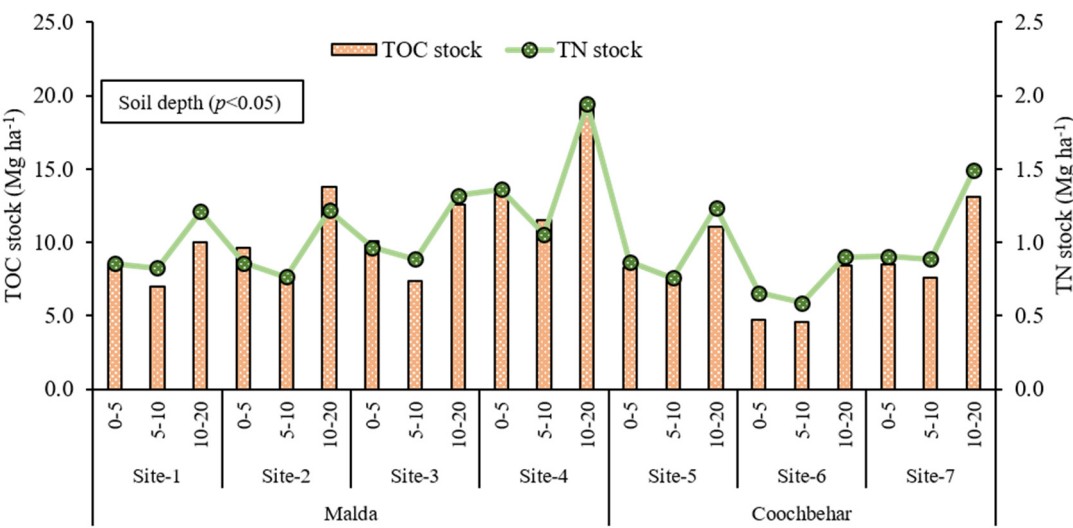

**Figure 5.** Effect of soil depths (0–5, 5–10, and 10–20 cm) on TOC and TN stocks at different sites of Malda and Coochbehar.

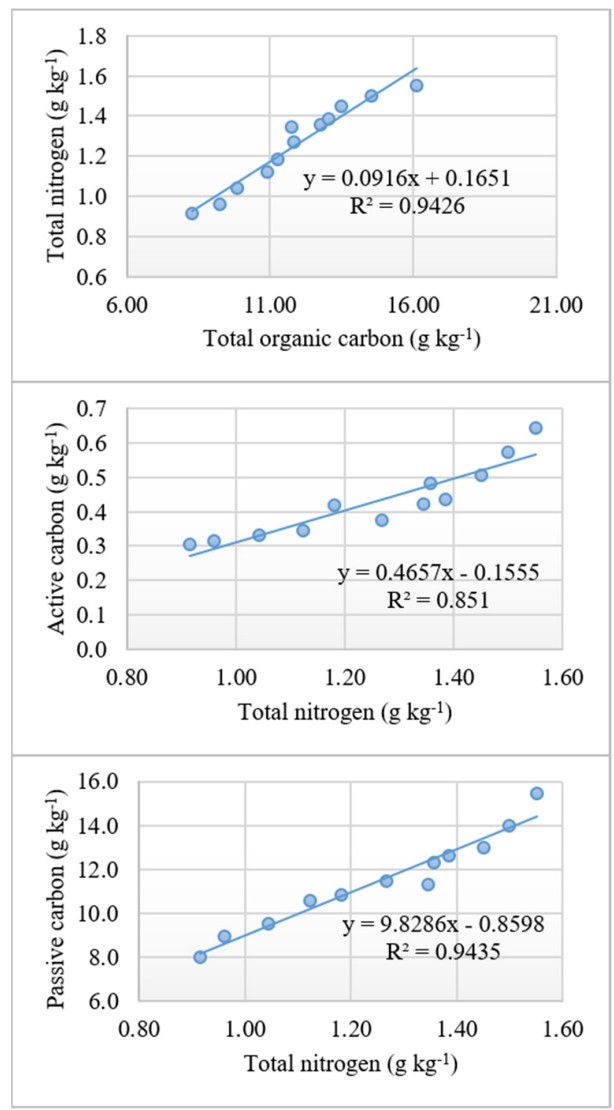

**Figure 6.** Relationship of TOC, TN, AC, and PC.

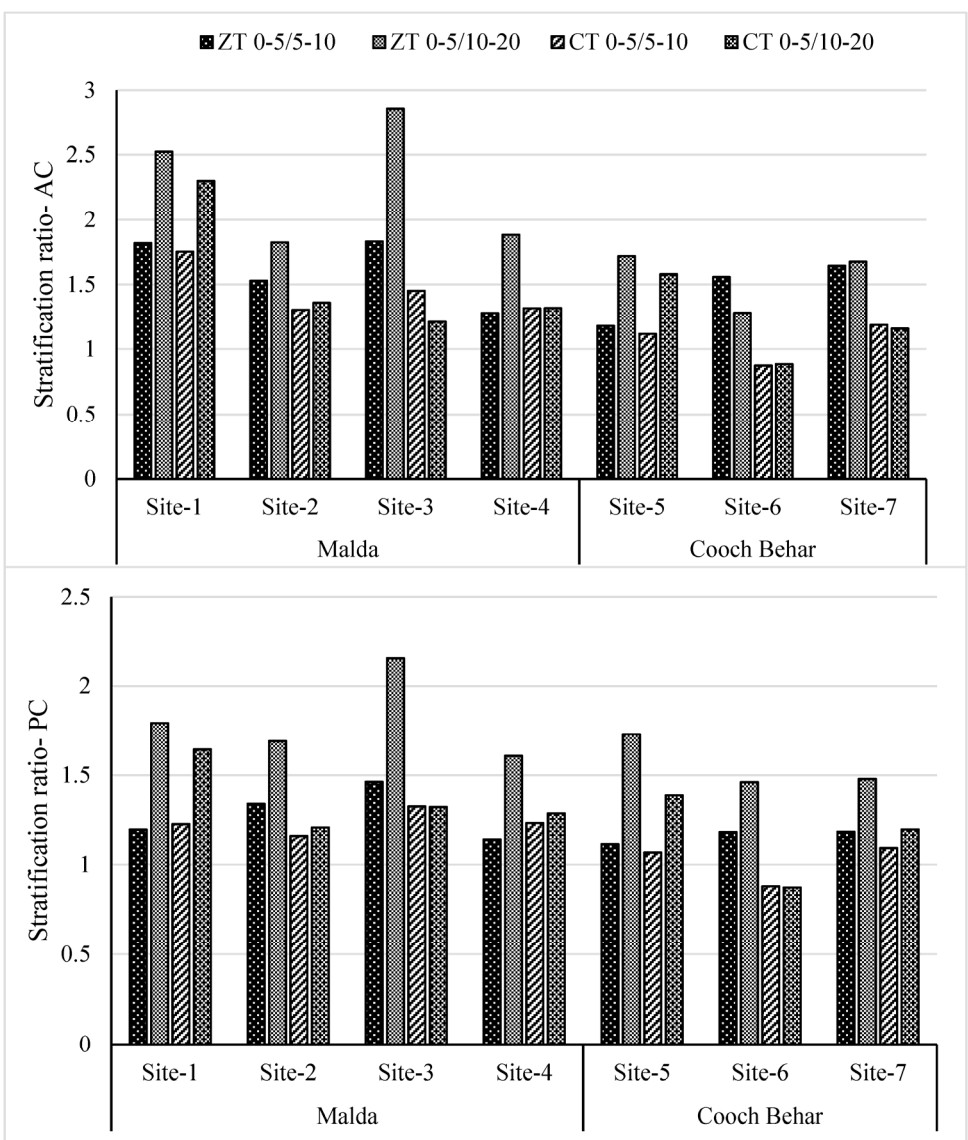

**Figure 7.** Stratification ratio of AC and PC at 0–20 cm soil as influenced by tillage practices.

### 3.6. Contribution of AC and PC to TOC and the Ratio of AC/PC

PC contributed the highest to the TOC (Figure 8) which varied from 95.6 to 97.2%, while the contribution of AC varied from 2.7 to 4.3%. We have observed that AC contribution was much higher in the Malda sites, whereas PC contribution was comparatively higher in the Coochbehar sites. We have noticed almost similar AC/PC ratio values with respect to tillage (Figure 9a) and cropping system (Figure 9b). However, there were some irregular patterns of ratio values also witnessed among the sites in this study.

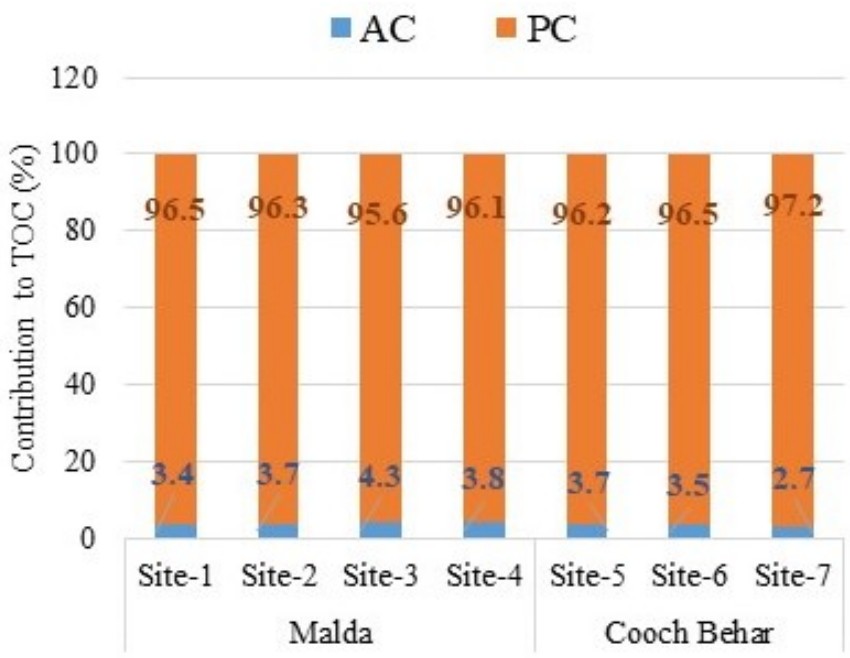

**Figure 8.** Percent contribution of AC and PC to TOC.

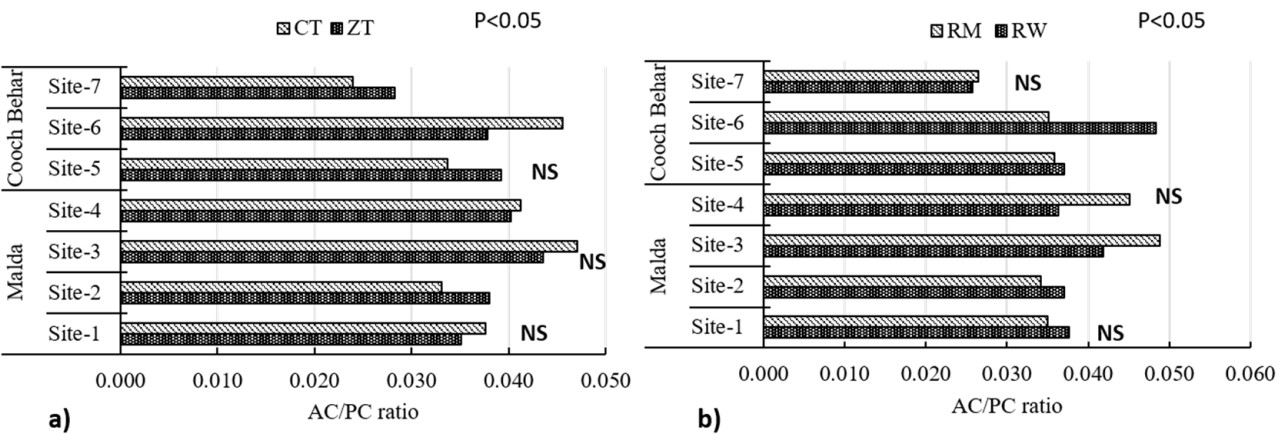

**Figure 9.** (**a**) ratio of AC/PC under tillage practices; (**b**) ratio of AC/PC under cropping systems. NS: Non-significant.

## 4. Discussion

### 4.1. Effect of Cropping System on C Fractions

In the present short-term investigation, the RM system was observed to be a dominant cropping system in enhancing the concentration of TOC and its fractions (AC and PC) which is also reflected in the TOC and TN stocks. In general, maize is a high-biomass crop compared to wheat. As a result, high crop biomass residue will increase the soil profile TOC storage which ultimately increases the different labile and non-labile C fractions. Many researchers also suggested that C fractions increased with the increase in organic matter (OM) input [58–62]. Recently, Bongiorno et al. [63] revealed in their study that all labile C fractions were significantly higher in high OM in comparison with low OM input treatments. Microbial biomass uses OM as its energy substance which is the key reason for the C pool increment in the soil system. The addition of organic substrates stimulates the microbial population which directly contributes to an increase in the labile organic carbon pools [64]. Different cropping systems have a substantial effect on active carbon as it is highly sensitive to management changes [65–67]. Besides the higher C input, the quality of substrate input is also important for decomposition [68]. Substrate quality is assessed

as the ratio of neutral detergent fiber (NDF-cellulose, hemicellulose, and lignin) to crude protein (CP) which is directly linked with the decomposition [69]. The soluble fraction present in wheat and maize crop biomass residues also influences the mineralization rate of C in the early stage of decomposition [70].

## 4.2. Effect of Tillage on C Fractions

The ZT resulted in significant improvements in TOC and its fractions and TN stocks against the CT. The influence of the ZT system on TOC to a greater level was reported by Carbonell-Bojollo et al. [71], Yadav et al. [72] and Rakesh et al. [21]. Several researchers also reported the positive effects of ZT on TOC against the CT system [73,74]. Under the effect of CT, we noticed a lesser amount of TOC ascribed due to the loss (through mineralization) that occurred during the tillage operation. Mineralization of SOC will be faster in CT plots because the plowed soil layers are associated with exposure from inversion and aggregate breakdown [73]. However, the excess addition of any crop residue (C biomass) decreases the oxidation rates and decomposition losses of C which ultimately helps increase the C storage in the soil profile [75,76]. Interestingly, we observed an enrichment of the upper surface layer (0–10 cm) with the C fractions under the effect of ZT, but in the lower soil depth (10–20 cm), CT enhanced the same. Higher microbial population and respiration activity at the topmost soil surface helps in the mineralization of C more quickly and conversion into protected TOC [77]. A similar increment in SOC at the soil surface (0–10 cm) under ZT was reported by Angers and Eriksen-Hamel [78] and Rakesh et al. [21]; an increment in the lower layer under CT was reported by Zhu et al. [79]. In CT, there is a direct effect of tillage where inversion occurs which is the main cause of the enhancement of SOC in deeper layers. Nevertheless, soil aggregation stability also plays a vital role in C storage. Tillage disturbance would increase the macroaggregate and reduce the microaggregate formation which is important for C stabilization [80]. However, Devine et al. [81] in their study reported that under the CT system, both the small macroaggregates and microaggregates helped in enhancing the total SOC compared to the large macroaggregates at the soil depth of 5–15 cm.

## 4.3. Interaction Effect of Tillage and Cropping System on C Fractions

Combining crop residues with tillage, i.e., CS × T resulted in a significant improvement in TOC and its fractions. In our study, we noted that ZT under maize cropping recorded a higher proportion of C accumulation along with TN as compared to CT. There were similar results under RW but comparatively fewer than RM. High biomass in maize crops is the key reason for higher C accumulation in comparison with wheat. Several researchers also revealed the effect of high residue biomass addition on SOC under the ZT. For example, cover crops have high C biomass, which can effectively increase SOC storage when used under the ZT practice. Increased SOC accumulation was observed when the cover crops were integrated into no-till systems [82–84]. The prominent effect of higher residue inputs combined with no-tillage on SOC was evident [67]. Jat et al. [62] in their study, revealed that after 6 years of experimentation, TOC was observed to be maximum in the maize-wheat system than the rice-wheat system under ZT conditions; this was attributed to 10 t ha$^{-1}$ more biomass residue added in the maize-wheat than the rice-wheat system. This indicates the importance of high C biomass residues in enhancing TOC under ZT. However, in some of the sites, CT also recorded maximum values that may be ascribed to the fact that the amount of residue addition in ZT and CT may vary with the farmers/growers and their culture of management practice.

## 4.4. Effect of Tillage and Cropping System on TOC and TN Stock

All the fractions along with TOC and TN concentration resulted in a decreased status with increasing soil depth, which is due to the natural stratification by residue accumulation on the soil surface; this was in agreement with de Moraes Sá and Lal [14]. Similar results were also noticed by Duan et al. [85]. However, the TOC and TN stocks observed to be

increasing with an increase in soil depth were evident in our study due to the textural difference and bulk density among the different soil types. Additionally, the soil sampling depths also influenced this variation as the 10–20 cm covered a high volume of soil compared to the above layers (0–5 and 5–10 cm). As a result, we observed higher stock values in lower soil depth. In our study, ZT practice and RM system with high biomass showed high values of TOC and TN stock. Maximum values of TOC stock under ZT were also reported by Alvarez et al. [86] and Metay et al. [87]. However, the actual changes in SOC stock are determined by the time period of ZT adoption [88]. In some cases, we noticed CT improved the stocks which are due to the inversing effect of crop residues during tillage [89].

### 4.5. Relationship of AC, PC, TOC, and TN

We observed a strong relationship between TOC and TN; additionally, AC and PC also positively correlated to TN indicating the importance of TN in enhancing the C pools and its stocks in the soil profile. Other researchers such as Holden and Treseder [90], Eze et al. [91], Liu et al. [92], and Duan et al. [85] also recorded a significant effect of active carbon and nitrogen on SOC and soil nitrogen stock and cycling. TOC is very important for nitrogen stock, as the nitrogen stores rapidly in SOM [93]. A significant positive relationship between soil organic carbon and active carbon has been widely reported [54,94–96]. Improved land use practices enhance the TOC which ultimately improves the status of C fractions in TOC; however, it also depended on the climate and soil type. A similar significant interrelationship between the TOC pools was also reported in the soil of different agro-ecoregions [30].

### 4.6. Effect of Tillage on Stratification of AC and PC

A higher stratification ratio of AC and PC values under the ZT system indicated its maximum accumulation in upper layers as compared to CT. Melero et al. [97] also reported similar higher stratification ratio values of AC under ZT. We observed lower SR values under CT because of the incorporation effect during tillage that allowed the distribution of SOC into lower profiles. The concentration of accumulation of C fractions was higher between 0–10 cm than at 0–20 cm. Lower stratification ratio values in Coochbehar soils indicated a higher profile distribution of SOC in the soil profile as the soils were sandy loam in texture. This result could also be caused by the high rainfall together with the sandy soil texture resulting in high runoff and deep percolations in Coochbehar. The old alluvial Inceptisol (Malda) showed a comparatively greater amount of SOC and its fractions in comparison with the new alluvial Entisol (Coochbehar). The concentration of AC and PC was observed to decrease with soil depth in both soils. Accumulation of the C fractions on the soil surface was greater in the Inceptisol because of heavy texture (high bulk density) and also the low frequency of rainfall resulted in higher stratification values. Moreover, in the clay soils of Malda, leaching losses are restricted because of the formation of clay soil organic matter complex and soils' lower hydraulic conductivity [98] which may further facilitate the C storage in soil. The Entisol soil type was lighter in texture which resulted in lower stratification values indicating a good distribution of C fractions in the soil profile. Zhang et al. [99] showed in their study that growing maize crops in black soil (36% clay, 24% silt, and 40% sand) under ZT resulted in greater accumulation in SOC.

### 4.7. Overall Effect of Conservation Agriculture among the Sites

Site three and site four of Malda district were rich in native SOC status (Table S1) and coincided with a management practice resulting in higher values of TOC, TN, AC, and PC. In most of the cases, the least amounts of C fractions were recorded in site six and site seven of Coochbehar because of the low native SOC status with poor management practice. The clay texture in site four further enhanced the accumulation rates of C fractions which indicated the role of clay particles in stocking the carbon. In some of the sites, we also noticed that CT improved C fractions better than ZT which can be explained by the variations in residue addition by the farmers; this naturally resulted in such disparities

in the data. The significant variability of C fractions in Coochbehar and Malda sites was observed due to the varying agricultural practices and site-specific conditions where the initial TOC status in the Malda soils was higher than in the Coochbehar soils. Alvarez [86] revealed that the variations in the relative effects of tillage on SOC could not be explained by the SOC content, temperature, precipitation, or soil texture. This result suggests that change in SOC under ZT will increase with increasing time.

*4.8. Contribution of AC and PC to TOC and Their Ratio in Soil*

In the present study, the contribution of PC fractions to the TOC was recorded to be greater than the AC among all the sites (Figure 8). In fact, AC is the chief source of nutrients and can be easily attacked by soil microbes, thus the concentration of AC reduces quickly as compared to PC. However, PC is very resistant to microbial attack and protected as organic-mineral complexes, and of difficult access [100], which may increase its relative proportion within the total SOC [101]. Because of the better soil type which influenced the decomposition rates in Malda sites resulted in higher AC turnover, while in Coochbehar, the acidic nature of the soil coupled with sandy texture and high rainfall, led to decomposition rates that were lower and partial; as a result, there was an increased loss of AC from soil and predominance of PC in Coochbehar. Furthermore, high clay in Malda sites favored the enhancement of AC. Clay content helps in preserving SOC [102] as organic mineral complexes protect labile C that can be easily attacked by soil enzymes in soil [103]. An almost similar type of ratio values within the treatments and sites of AC/PC in our study indicated that changes in tillage or cropping systems do not affect the AC/PC ratio greatly. A similar type of results indicating no significant changes in the ratio of carbon fractions under soil management practices was also reported by Liu et al. [104] and Yu et al. [105].

**5. Conclusions**

Our study systematically demonstrated the differences in storing the AC and PC fractions among different climate and soil types which is helpful for future studies that focus on carbon management under different land use management and climate effects. The status of AC and PC fractions was observed to be parallel with the addition of high substrate carbon which reflected from the higher biomass addition under maize cropping. TOC and TN stocks enhanced substantially under RM as compared to RW. Adoption of ZT improvised the status of C fractions at top layers, while CT improved the same in the lower soil depths because of the inversion effect during tillage. Interactively, ZT performed better with the prominent effect of higher residue inputs under the RM system for enhancing the C fractions in the soil profile. TOC and TN stock increase with soil depth was evident due to the textural and bulk density difference among the two different soil types. Along with TOC, the AC and PC also positively regressed on TN indicating the importance of TN in enhancing the C stocks in the soil profile. A higher stratification ratio of AC and PC under the ZT system indicated its maximum accumulation in upper layers as compared to CT. Lower stratification ratio values in Coochbehar soils (new alluvial Entisol) indicated higher profile distribution of SOC in soil profile due to sandy loam texture. A greater accumulation of the C fractions on the soil surface of the Inceptisol was due to the heavy texture that resulted in higher stratification values. High rainfall coupled with acidic soil reaction in the new alluvial Entisol resulted in partial decomposition of crop residue biomass, which increased the macroaggregate formation with sand and resulted in less C sequestration in these soils.

The novelty of the present investigation is in demonstrating the variations in AC and PC accumulation in two soil types that resulted in the potentiality of Inceptisol types in stocking the maximum amount of carbon and the potentiality of Entisol types in distributing the same in the soil profile under the effect of two contrasting tillage and cropping types. Implementation of conservation agriculture with higher biomass additive crops would improvise the profile storage of C fractions as compared to low biomass inputs. The present study suggests considering the actual amount of root and shoot biomass C input to

authentically assess the accumulation patterns of C fractions under varying soil types and climates for future studies.

**Supplementary Materials:** The following supporting information can be downloaded at: https://www.mdpi.com/article/10.3390/land12020365/s1, Table S1: Soil pH, total organic C, total N, texture and bulk density (0–20 cm) of the experimental sites; Table S2: Interaction effects of cropping system, tillage and soil depths on total organic C (TOC) (g/kg) at different sites of Malda and Coochbehar districts; Table S3: Interaction effects of cropping system, tillage and soil depths on total nitrogen (TN) (g/kg) at different sites of Malda and Coochbehar districts; Table S4: Interaction effects of cropping system, tillage and soil depths on active carbon (AC) (g/kg) at different sites of Malda and Coochbehar districts; Table S5: Interaction effects of cropping system, tillage and soil depths on passive carbon (PC) (g/kg) at different sites of Malda and Coochbehar districts.

**Author Contributions:** Conceptualization, S.R. and A.K.S.; methodology, S.R. and A.K.S.; validation, S.R., A.K.S. and D.S.; formal analysis, S.R. and D.S.; investigation, S.R. and A.K.S.; data curation, D.R., D.B. and S.S.; software, S.R., P.K.J., P.K.D. and S.S.; writing—original draft preparation, S.R. and A.K.S.; writing—review and editing, S.R., D.S., A.K.S., A.R., P.K.J., P.K.D., D.R. and D.B.; visualization, S.R., A.K.S. and A.R.; supervision, A.K.S. All authors have read and agreed to the published version of the manuscript.

**Funding:** This research received no external funding.

**Data Availability Statement:** Data is contained within the article or Supplementary Material.

**Acknowledgments:** Authors are thankful to the Australian Centre for International Agricultural Research (ACIAR)—Sustainable and Resilient Farming System Intensification (SRFSI) team, International Maize and Wheat Improvement Center (CIMMYT), Bangladesh, and Department of Soil Science and Agricultural Chemistry, Uttar Banga Krishi Viswavidyalaya (UBKV), Pundibari, Coochbehar, West Bengal for providing the support in sampling and laboratory facilities. The authors extend their sincere thanks to the project-associated farmers of field trials and field technicians of both Malda and Coochbehar districts for helping in the collection of samples.

**Conflicts of Interest:** The authors declare no conflict of interest.

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
