# Peer review of "Active and Passive Carbon Fractions in Contrasting Cropping Systems, Tillage Practices, and Soil Types"

_land, doi:10.3390/land12020365_

Round 1

Reviewer 1 Report

my comments are in attachment !

Author Response

Reviewers’ and editors’ comments and suggestions

Authors’ responses (Line number in the revised manuscript)

This manuscript aims to quantify carbon storage potential of different soils under different cropping system and tillage practices.

The study is important and reveals information about how specific cropping system and tillage practices can be adjusted to maximize the carbon storage.

Thanks for your positive opinion

Overall, authors need to revise English language of the manuscript

English corrections have been made with care.

 L40-43: please check the font size

Font size corrected

How soil sampling points were identified? How many soil samples were collected? It is important to ensure that field variability was taken into account.

Sampling details and field variability details have been incorporated in the Material Section as suggested.

No information, how crop fields were managed such as irrigation, fertilization or any other relevant management practices

Information on field operation have been included under the separate heading 2.3. Crop Management

I would suggest to include more detailed information about monthly mean rainfall and temperature, this can help to discuss differences among the sites

Daily rainfall and temperature details have been presented in the new Figure 1. Also discussed accordingly in the Discussion part.

Authors did not reflect upon the influence of high or low AC and PC on overall soil productivity, what’s the ideal ratio to maintain soil productivity in the long term, addition of such discussion would be great

Thank you so much for your valuable suggestion. We have estimated the AC/PC ratio and their contribution newly and presented in figure 8 and figure 9. Also included in the result section and discussed accordingly.

L231-32- Please correct “under varied cropping systems”

Corrected as suggested

Information about the previous cropping pattern on these fields under experiment would add value

Information about the previous cropping has mentioned in the Material section.

L276: “under tillage” is not correct, can be replaced with under two tested tillage practices

Replaced with “under two tested tillage practices”

In discussion, authors talk about crop residues but no mention in materials and methods, was crop residues handled in standard way or most of it was left on field? Need to state that clearly in materials and methods. In discussion authors mention that high maize biomass yield was the main reason for higher C accumulation, was whole biomass as crop residue left on field? And how it was handled especially in case of ZT.

A brief detail on the crop residue management in both the cropping systems have been detailed under the 2.3. Crop Management Section.

Also detailed how the crop residue management was handled in ZT and CT.

The discussion chapter is very superficial, authors need to focus on discussing own results, trends and variation rather than summarizing results and putting references. Everything in discussion was around residue, the explanation about trends and variations across sites goes beyond that. For examples, authors must take into account the original soil data collected at the beginning of the experiment and take into account those differences among sites for discussion. For mineralization, temperature and rainfall are crucial thus authors need to include that in discussion as well.

Discussion part is now thoroughly revised in the light of the comments received by both the reviewers.

Separate sections are made and connected the results with appropriate references.

Connected the effect of climate factors to the stabilization of C.

Effect of initial soil discussed under the section 4.7. Overall effect of conservation agriculture among the sites

Reviewer 2 Report

I have reviewed the manuscript titled " Active and passive carbon fractions in contrasting cropping systems, tillage and soil types". This manuscript discusses the active and passive Carbon sequestration of cropping system with soil types.

I do find it suitable for the Land but I have the following observations on this MS. 

The introduction is weak, and the method section is trivial and vague at places. More recent literature work is required.

I don't feel qualified to judge the English language and style but the English language needs improvement.

Overall, the dataset used in this research was conduct through field surveys and all the data was generated by the author’s which is most reliable but the analysis conducted into the research paper is mostly simple and old. Therefore, I recommend the author to improve the analysis part of this paper.

The MS does not contribute new in terms of methodology - a set of well-known methods have been applied in recent studies. These methods are important as well but author needs to refine this methodology and be more specific for scientific problem.

I fail to see any fruitful discussion on the generated datasets. The introduction must be improved and the scientific problem has to be clearly identified and addressed.

I do see little novelty in both scientific findings but it needs improvement. First, the authors should clearly state the scientific significance of carbon data collected from field, rather than saying something very broad.

In Discussion, "Authors should discuss the results and how they can be interpreted in perspective of previous studies and the working hypotheses in more precise way.

Author Response

Reviewers’ and editors’ comments and suggestions

Authors’ responses (Line number in the revised manuscript)

I have reviewed the manuscript titled " Active and passive carbon fractions in contrasting cropping systems, tillage and soil types". This manuscript discusses the active and passive Carbon sequestration of cropping system with soil types.

Thanks for understanding our research study.

I do find it suitable for the Land but I have the following observations on this MS. 

Thanks for your positive feedback.

The introduction is weak, and the method section is trivial and vague at places. More recent literature work is required

Introduction and Methods sections are now thoroughly revised as shown in track change.

I don't feel qualified to judge the English language and style but the English language needs improvement.

English corrections have been made with due care.

Overall, the dataset used in this research was conduct through field surveys and all the data was generated by the author’s which is most reliable but the analysis conducted into the research paper is mostly simple and old. Therefore, I recommend the author to improve the analysis part of this paper.

Thanks for your opinion. We are now clarified all the methodologies in detail to make the readers understand our hypothesis and analysis.

Please see the same in track changes.

The MS does not contribute new in terms of methodology - a set of well-known methods have been applied in recent studies. These methods are important as well but author needs to refine this methodology and be more specific for scientific problem.

We have revised the Methodology section critically for the transparent understanding of our methods and its implementation.

New thing in our methodology- is placing the varying amount crop residues as mulch and incorporated under different cropping systems and tillage practices. These effects are potentially studied in the three soil depths to understand the capacity of soil profile layer-wise accumulation of C fractions. Additionally, these variations have been compared among different climates at Indo-Gangetic plains.

The introduction must be improved and the scientific problem has to be clearly identified and addressed.

The introduction part has been improved a lot as suggested.

I do see little novelty in both scientific findings but it needs improvement. First, the authors should clearly state the scientific significance of carbon data collected from field, rather than saying something very broad.

Novelty of the research have been highlighted and linked the hypothesis of our study in the revised paper. Please see in the track changes.

In Discussion, "Authors should discuss the results and how they can be interpreted in perspective of previous studies and the working hypotheses in more precise way.

Discussion part is now thoroughly revised in the light of the comments received by both the reviewers.

Round 2

Reviewer 1 Report

Authors have revised the manuscript sufficiently, therefore it can be accepted for publication after minor checks.

Reviewer 2 Report

I would like ton congratulate the authors for revising the manuscript and improving the quality of the paper.